

# *In vitro* assessment of the impact of 30 CYP2C19 variants on citalopram metabolism

Peng Wang[1,*], Xiao-xia Hu[2,*] and Jun-wei Li[3]

[1] Department of Clinical Pharmacy, Jinhua People's Hospital, Affiliated Jinhua Hospital of Wenzhou Medical University, Jinhua, Zhejiang, China
[2] Department of Clinical Pharmacy, Affiliated Jinhua Hospital, Zhejiang University School of Medicine, Jinhua, Zhejiang, China
[3] School of Pharmaceutical Sciences, Wenzhou Medical University, Wenzhou, ZheJiang, China
* These authors contributed equally to this work.

Corresponding author
Jun-wei Li, gravity172@sina.com

## ABSTRACT

**Background:** *CYP2C19* polymorphisms are correlated with individual variability in response to citalopram treatment. The pharmacogenomic testing of *CYP2C19* has been shown to optimize the safety and efficacy of citalopram medication. Exploration of the effect of novel CYP2C19 variants on citalopram could further enhance the potential for personalized citalopram treatment.

**Objectives:** The main goal of this study was to functionally characterize 30 CYP2C19 variants in citalopram metabolism, most of which were rare or novel variants identified in the Chinese Han population.

**Methods:** An *in vitro* incubation system was set up using recombinant human CYP2C19 variants expressed in Sf21 insect cell microsomes to simulate the citalopram metabolic environment. A high-performance liquid chromatography with fluorescence detection method (HPLC-FLD) was established to quantitatively determine both citalopram and demethylcitalopram.

**Results:** In this study, compared to the wild-type enzyme (CYP2C19*1), 73% (22/30) of the CYP2C19 variants showed significantly different metabolic activities in citalopram metabolism. Among them, two variants, CYP2C19*29 and L16F, showed significantly increased intrinsic clearance (nearly 5-fold and 1.5-fold, respectively). Eighteen variants–CYP2C19*2C, *2F, *2G, *2J, *6, *18, *30, *31, *32, *33, N231T, R124Q, R261W, I327T, A430V, R125G, M255T, and I331V–exhibited significantly decreased intrinsic clearance (18.02–63.16%). Two variants, CYP2C19*3 and 35FS, demonstrated weak or no activity. Moreover, the remaining 27% (8/30) of the CYP2C19 variants showed similar metabolic activities to that of the wild-type enzyme.

**Conclusion:** These CYP2C19 variants require specific attention from physicians and researchers, as their altered metabolic activities may influence the safety and efficacy of citalopram treatment. This work greatly expands the previously underexplored knowledge about the metabolic activities of rare or novel CYP2C19 variants in relation to citalopram medication. These findings may further facilitate the precision use of citalopram in personalized medicine.

# INTRODUCTION

Depressive disorders are a major public health concern worldwide, characterized by high incidence, high recurrence, and high disability rates, which puts a serious burden on families and society (*Global Burden of Disease Study 2013 Collaborators, 2015*; *Malhi & Mann, 2018*; *National Institute for Health and Care Excellence: Guidelines, 2022*). In particular, in China, depressive disorders have become an increasingly important public health priority due to the large population and the increasing incidence (*Meng et al., 2020*; *Tang, Jiang & Tang, 2022*; *Bai et al., 2022*). Antidepressant medication is an important clinical intervention that can decrease suicidal ideation along with other symptoms (*Malhi & Mann, 2018*; *Simon, Moise & Mohr, 2024*). However, there are substantial inter-individual discrepancies in the therapeutic response and adverse reactions of antidepressant medications, which may be related to gene polymorphisms (*Milosavljevic et al., 2021*; *Murphy et al., 2022*; *Bousman et al., 2023*). A growing body of research has reported that pharmacogenomic testing can improve clinical outcomes in antidepressant medication treatment (*Bahar et al., 2020*; *Murphy et al., 2022*). In order to guide clinical practice, the Clinical Pharmacogenetics Implementation Consortium (CPIC), a globally influential academic organization, has issued a guideline on using gene polymorphisms results to inform the prescribing of serotonin reuptake inhibitor antidepressants (*Bousman et al., 2023*). This guideline is widely recognized by medical and healthcare professionals to assist clinical decision-making and optimize healthcare policies. Therefore, precision medication therapy on the basis of individual genetic information is an inevitable trend to deal with the public health challenges posed by depression.

CYP2C19 is one of the most commonly studied members of the cytochrome P450 superfamily of enzymes. It exhibits functionally relevant polymorphisms, resulting in significant individual differences in response to antidepressant treatments. There is considerable evidence that *CYP2C19* pharmacogenetic tests can optimize antidepressant treatment by improving response rate and identifying potential adverse reactions (*Bousman et al., 2023*). However, the use of routine tests for the most common CYP2C19 variants has been shown to enhance the safety and efficacy of antidepressant medication in only 36% of patients (*Kee et al., 2023*). This indicates that many patients do not derive the benefits of antidepressant treatment, even after routine genotyping. The CPIC guideline also acknowledges in the limitations that selective *CYP2C19* allelic screening may miss novel or other clinically significant variants (*Bousman et al., 2023*). Meanwhile, a growing number of researchers suggest that integration of common, rare and individual variants could further enhance the potential for personalized antidepressant treatment (*Borczyk et al., 2022*). Therefore, it is essential to investigate the effects of *CYP2C19* genetic variants on antidepressant efficacy, particularly in diverse ethnic groups and for rare variants.

Citalopram is a first-line pharmacotherapy for depression, exerting antidepressant effects mainly through the inhibition of serotonin reuptake from the synaptic cleft

(*Simon, Moise & Mohr, 2024*). It is mainly metabolized in the liver to demethylcitalopram, with CYP2C19 being a key enzyme involved in the demethylation metabolism (*Brøsen & Naranjo, 2001*). Studies have shown that *CYP2C19* polymorphisms significantly affect citalopram exposure *in vivo* and the clinically relevant effect (*Chang et al., 2014*; *Zastrozhin et al., 2021*; *Wong et al., 2023*). Specifically, *CYP2C19* poor metabolizers exhibit elevated plasma concentrations of citalopram, which may increase the probability of dose-related side effects such as QT prolongation and Torsade de Pointes. Conversely, the *CYP2C19* ultrarapid metabolizers have lower plasma concentrations, which may lead to treatment failure. These differences in therapeutic efficacy and toxicity associated with *CYP2C19* polymorphisms have also been confirmed in the Chinese population (*Yin et al., 2006*; *Huang et al., 2021*). Therefore, it is crucial to investigate the impact of CYP2C19 variants, including rare and novel variants, on citalopram metabolism.

The distribution frequencies of *CYP2C19* polymorphisms exhibit considerable interethnic differences (*Zhou, Ingelman-Sundberg & Lauschke, 2017*). Currently, a plethora of studies have explored the effect of *CYP2C19* polymorphisms on citalopram metabolism, focusing mainly on these alleles with known functional properties, such as *CYP2C19*\*2, *3 and *17, in populations of European genetic ancestry. However, few studies have investigated the impact of critical or rare *CYP2C19* variants on citalopram metabolism in Chinese populations. In our previous study, we systematically investigated the genetic polymorphisms of the *CYP2C19* gene in the Chinese Han population by amplifying all nine exons in 2,127 unrelated healthy subjects using direct sequencing (*Hu et al., 2012*). In this population, we identified 30 variants that theoretically result in the substitution of amino acid residues, which may influence enzyme activity. Among them, 16 variants have been named (CYP2C19*2C, *2E–*2H, *2J, *3, *3C, *6, *18, *23, and *29–*33), and the remaining 14 variants are not well-studied and their functional impacts remain unknown (Table 1). The majority of these variants (28/30) were rare (frequencies typically below 0.1%), and have not been included in current CPIC guidelines. According to the PharmGKB database (*PharmGKB, 2025*), except for CYP2C19*3, which is classified as no-function variant, the functions of the remaining 29 variants are either uncertain or supported by limited evidence. Therefore, the current study systematically analyzes the enzymatic characteristics of 30 CYP2C19 variants toward citalopram metabolism and offers valuable information relevant for global pharmacogenomics research and clinical practice.

# MATERIALS AND METHODS

## Chemicals and materials

Citalopram (purity 98.0%) and venlafaxine (purity 98.0%, internal standard, IS) were obtained from Tokyo Chemical Industry Co., Ltd. (Tokyo, Japan). Demethylcitalopram (purity 98.0%) was purchased from Toronto Research Chemicals Inc. (TRC, Pickering, Ontario, Canada). Cytochrome b5 microsomes and recombinant human CYP2C19 (expressed in *Spodoptera frugiperda* (Sf) 21 insect cells microsomes) were provided by the Beijing Institute of Geriatrics, National Health Commission (Beijing, China). Reduced nicotinamide adenine dinucleotide phosphate (NADPH) was obtained from Sigma
**Table 1 Enzyme kinetic parameters of citalopram demethylation activity of recombinant CYP2C19 enzyme wild-type and 30 variants.**

| Variants | cDNA Change | Main effect | rs ID | Region | Naming status | Frequencies (%) | Vmax (pmol/min/ pmol P450) | Km (μM) | CLint (Vmax/ Km) (μL/min/nmol P450) | Relative clearance (% of wild type) |
|---|---|---|---|---|---|---|---|---|---|---|
| 2C19*1 | | | | | | | 19.10 ± 0.45 | 122.67 ± 9.67 | 156.14 ± 8.87 | 100.00 |
| 2C19*2C | 481G > C | A161P | rs181297724 | Exon 4 | Named | 0.05 | 5.39 ± 0.09* | 115.87 ± 8.28 | 46.67 ± 2.51* | 29.93* |
| 2C19*2E | 813G > A | M271I | rs778258371 | Exon 5 | Named | 0.02 | 12.84 ± 0.38* | 129.20 ± 20.15 | 100.85 ± 14.28 | 64.29 |
| 2C19*2F | 1021G > A | D341N | rs770829708 | Exon 7 | Named | 0.05 | 8.86 ± 0.10* | 96.92 ± 3.23 | 91.54 ± 4.15* | 58.79* |
| 2C19*2G | 1079A > T | D360V | rs550527959 | Exon 7 | Named | 0.05 | 13.19 ± 0.13* | 147.30 ± 8.61 | 89.70 ± 4.51* | 57.59* |
| 2C19*2H | 1186C > G | H396D | rs1564686367 | Exon 8 | Named | 0.02 | 16.85 ± 0.28 | 138.83 ± 9.38 | 121.78 ± 9.42 | 78.15 |
| 2C19*2J | 1261A > C | K421Q | / | Exon 8 | Named | 0.02 | 12.99 ± 0.09* | 154.13 ± 10.31 | 84.50 ± 5.49* | 54.14* |
| 2C19*3 | 636G > A | W212X | rs4986893 | Exon 4 | Named | 5.34 | ND | ND | ND | ND |
| 2C19*3C | 407T > A | M136K | rs763625282 | Exon 3 | Named | 0.07 | 21.36 ± 0.57 | 171.10 ± 13.05 | 125.15 ± 6.68 | 80.14 |
| 2C19*6 | 395G > A | R132Q | rs72552267 | Exon 3 | Named | 0.09 | 4.23 ± 0.14* | 151.33 ± 17.61 | 28.13 ± 2.42* | 18.06* |
| 2C19*18 | 986G > A | R329H | rs138142612 | Exon 7 | Named | 0.02 | 6.45 ± 0.00* | 122.97 ± 1.10 | 52.43 ± 0.47* | 33.64* |
| 2C19*23 | 271G > C; | G91R | rs118203756 | Exon 2 | Named | 0.05 | 23.81 ± 0.77* | 126.10 ± 8.11 | 189.05 ± 5.97 | 121.15 |
| 2C19*29 | 83A > T | K28I | rs1564656981 | Exon 1 | Named | 0.02 | 46.31 ± 0.59* | 60.39 ± 2.57 | 767.55 ± 23.24* | 492.31* |
| 2C19*30 | 217C > T | R73C | rs145328984 | Exon 2 | Named | 0.02 | 7.48 ± 0.17* | 132.50 ± 9.51 | 56.55 ± 2.76* | 36.23* |
| 2C19*31 | 232C > T | H78Y | rs1564660997 | Exon 2 | Named | 0.02 | 10.25 ± 0.15* | 152.93 ± 2.17 | 67.02 ± 0.08* | 43.03* |
| 2C19*32 | 296A > G | H99R | rs1288601658 | Exon 2 | Named | 0.02 | 11.43 ± 0.14* | 116.27 ± 7.37 | 98.62 ± 7.42* | 63.30* |
| 2C19*33 | 562G > A | D188N | rs370803989 | Exon 4 | Named | 0.02 | 10.98 ± 0.21* | 131.93 ± 0.99 | 83.21 ± 2.16* | 53.43* |
| 35FS | 101-102insCCTAC | 35 frameshift | / | Exon 1 | Novel | 0.02 | ND | ND | ND | ND |
| N231T | 692A > C | N231T | / | Exon 5 | Novel | 0.02 | 11.37 ± 0.22* | 137.33 ± 5.00 | 82.92 ± 4.40* | 53.24* |
| R124Q | 371G > A | R124Q | rs200346442 | Exon 3 | Novel | 0.02 | 9.81 ± 0.11* | 100.13 ± 9.17 | 98.50 ± 8.03* | 63.10* |
| R261W | 781C > T | R261W | / | Exon 5 | Novel | 0.02 | 7.62 ± 0.29* | 110.10 ± 4.62 | 69.32 ± 3.83* | 44.48* |
| S303N | 908G > A | S303N | / | Exon 6 | Novel | 0.02 | 13.16 ± 0.12* | 70.24 ± 2.22 | 187.45 ± 4.46 | 120.28 |
| I327T | 980T > C | I327T | / | Exon 7 | Novel | 0.02 | 6.48 ± 0.04* | 112.00 ± 6.06 | 57.93 ± 2.84* | 37.18* |
| A430V | 1289C > T | A430V | / | Exon 8 | Novel | 0.02 | 9.85 ± 0.02* | 113.90 ± 4.36 | 86.52 ± 3.13* | 55.54* |
| R125G | 373C > G | R125G | / | Exon 3 | Novel | 0.02 | 10.65 ± 0.47* | 178.40 ± 22.07 | 60.07 ± 4.74* | 38.46* |
| N277K | 831C > A | N277K | / | Exon 6 | Novel | 0.07 | 18.46 ± 0.23 | 119.30 ± 6.38 | 154.96 ± 6.38 | 99.41 |
| N403I | 1208A > T | N403I | / | Exon 8 | Novel | 0.05 | 8.03 ± 0.07* | 76.28 ± 1.65 | 105.32 ± 3.14 | 67.60 |
| M255T | 764T > C | M255T | / | Exon 5 | Novel | 0.02 | 10.90 ± 0.12* | 159.63 ± 3.90 | 68.31 ± 1.81* | 43.87* |
| T130M | 389C > T | T130M | rs150152656 | Exon 3 | Novel | 0.05 | 35.47 ± 0.47* | 135.77 ± 11.96 | 262.58 ± 22.41 | 168.10 |
| L16F | 46C > T | L16F | / | Exon 1 | Novel | 0.02 | 44.29 ± 2.33* | 184.90 ± 4.11 | 239.45 ± 7.46* | 153.62* |
| I331V | 991A > G | I331V | rs3758581 | Exon 7 | Novel | 90.16 | 7.79 ± 0.14* | 91.68 ± 7.30 | 85.27 ± 5.47* | 54.68* |

**Notes:**
* $P < 0.05$ (*vs.* wild-type).
"Named" refers to variants officially designated as star alleles in the PharmVar database; "Novel" indicates newly identified variants not yet named by PharmVar; "ND" indicates not determined. For named variants, only the main substitution was analyzed.

(St. Louis, MO, USA). All other chemicals and solvents used were of analytical grade and were obtained from Beijing Chemical Factory Co., Ltd. (Beijing, China).

### *In vitro* enzymatic activity assay

According to the previously reported method (*Dai et al., 2015*), all of the CYP2C19 variants were amplified by overlap extension polymerase chain reaction (PCR) using

wild-type cDNA as the template. The mutant sequences were ligated into the dual-expression baculovirus vector (pFastBac-OR-CYP2C19) and verified by gene sequencing. After infection of Sf21 cells, protein levels were measured by immunoblotting, and the concentration of CYP2C19 active holoproteins in microsomal proteins was determined by reduced carbon monoxide difference spectroscopy. The total volume of the enzyme catalytic reaction system was 200 µL. The system contained 5 pmol recombinant human CYP2C19 enzyme (wild-type or 30 other variants), 5 pmol cytochrome b5, a gradient of citalopram concentration (10–1,000 µM), and 0.1 M PBS buffer (pH 7.4). After thorough mixing, the reaction mixture was pre-incubated at 37 °C for 5 min. The reaction was initiated by adding 10 µL NADPH solution (20 mM), and the mixture was incubated at 37 °C for 30 min. The reaction was terminated by transferring the mixture to −80 °C for 15 min. After retrieving the sample from −80 °C, 50 µL of 0.1 M hydrochloric acid, 0.8 mL of ethyl acetate, and 30 µL of 10 µg/mL venlafaxine solution (internal standard) were added. The sample was thawed and vortexed for 2 min, followed by centrifugation at 12,000 r/min for 10 min at 4 °C. 0.7 mL of supernatant was collected and evaporated under nitrogen. The residue was redissolved in the mobile phase, and a 2 µL aliquot was used for quantitative determination. Three parallel samples were prepared for each variant at each citalopram concentration (10–1,000 µM).

## Chromatographic conditions and method

This study established an high-performance liquid chromatography with fluorescence detection method (HPLC-FLD) method to quantitatively determine both citalopram and demethylcitalopram quickly and accurately. An Agilent 1260 HPLC instrument coupled to an Agilent 1260 FLD Spectra (G1321B) fluorescence detector (Santa Clara, CA, USA) was used in the study. An Agilent RRHD Eclipse Plus C18 column (3.0*100 mm, 1.8 µm) was used for separation. The mobile phase consisted of acetonitrile (A) and 0.05% trifluoroacetic acid (B). The flow rate was controlled at 0.3 mL/min. The elution occurred in gradient mode with the following conditions: 30–28% (A) and 70–72% (B) in the first 5 min; 28–30% (A) and 72–70% (B) between 5 and 9 min; and 30% (A) and 70% (B) maintained for 2 min. The column temperature was maintained at 30 °C. The excitation and emission wavelengths were set at 245 nm and 306 nm, respectively. The total run time was 11 min.

## Statistical analysis

The Michaelis-Menten curves for CYP2C19 variants were fitted with citalopram concentration (X-axis) *versus* demethylcitalopram formation rate (Y-axis) using GraphPad Prism 6 (GraphPad Software, La Jolla, CA, USA). The Michaelis constant (Km) and maximum velocity (Vmax) values were calculated by curve fitting. Intrinsic clearance (CLint) was determined as the Vmax/Km ratio. Differences in the metabolic parameters of citalopram between the wild-type enzyme and 30 other variants were analyzed using Dunnett's T3 multiple comparison in SPSS Statistics version 24 software (IBM Corporation, Armonk, NY, USA). The data were presented as mean±standard deviation. Statistical significance was denoted at $P < 0.05$.

## RESULTS

The HPLC-FLD method was validated according to conventional requirements. As shown in Fig.1, the retention times of venlafaxine, demethylcitalopram, and citalopram were 3.8, 7.2, and 8.1 min, respectively. The analytes were well separated with no interference. The linear regression curve for demethylcitalopram was established using the ratio of peak area to concentration. The linear concentration range was determined to be 50 to 2,500 ng/mL, with a coefficient of determination of 0.9998, which indicated excellent linearity. The lower limit of quantification for demethylcitalopram was 50 ng/mL. The precision, accuracy, recovery, matrix effect, and stability of the method were adequate.

The demethylation activity of the CYP2C19 wild-type enzyme and 30 other variants toward citalopram were evaluated in this experiment. The fitted Michaelis-Menten curves for all CYP2C19 variants are presented in Fig.2, and the corresponding enzyme kinetic parameters are summarized in Table 1. The results indicated that demethylcitalopram was not detected in the CYP2C19*3 and 35FS enzyme incubation systems, so their kinetic parameters could not be evaluated. This suggested that they had completely lost catalytic activity. Furthermore, the kinetic parameters of almost all CYP2C19 variants were significantly altered compared to the wild-type. Specifically, the Vmax value of four variants (CYP2C19*23, *29, T130M, and L16F) was significantly increased, achieving 1.25- to 2.42-fold that of the wild-type, whereas the Vmax of three variants (CYP2C19*2H, *3C, and N277K) did not differ significantly from the wild-type. The remaining 21 variants exhibited significantly decreased Vmax, ranging from 22.15% to 69.06% of the wild-type. The Km value showed no significant difference among the variants.

The CLint was used as an index to evaluate the demethylation metabolism of the CYP2C19 enzyme toward citalopram in our experiment. The relative clearance rate was used to display differences in citalopram metabolic capacity among CYP2C19 variants and the wild-type. It was expressed as the ratio of the CLint of each variant to that of the wild-type, as shown in Fig. 3. The CYP2C19*29 and L16F variants showed significant increases in CLint, nearly 5-fold and 1.5-fold, respectively, compared to the wild-type. Meanwhile, the CLint of eighteen variants (CYP2C19*2C, *2F, *2G, *2J, *6, *18, *30, *31, *32, *33, N231T, R124Q, R261W, I327T, A430V, R125G, M255T, and I331V) was significantly lower than that of the wild-type, ranging from 18.02% to 63.16%. The remaining eight variants showed no significant difference in CLint.

The 30 variants were manually classified into six groups according to the degree of change in CLint compared to the wild-type. The CYP2C19*3 and 35FS variants, which had no enzymatic activity, were classified as the poor metabolism variants. Nine variants (CYP2C19*2C, *6, *18, *30, *31, R261W, I327T, R125G, M255T) exhibited 10–50% of the wild-type CLint and were regarded as the intermediate metabolism variants. Another nine variants (CYP2C19*2F, *2G, *2J, *32, *33, N231T, R124Q, A430V, I331V) showed 50–70% of wild-type and were regarded as the mild reduction variants. The L16F variant, which showed a 130% to 200% increase compared to wild-type, was regarded as the rapid metabolism variant, while the CYP2C19*29 variant, with more than a 200% increase, was

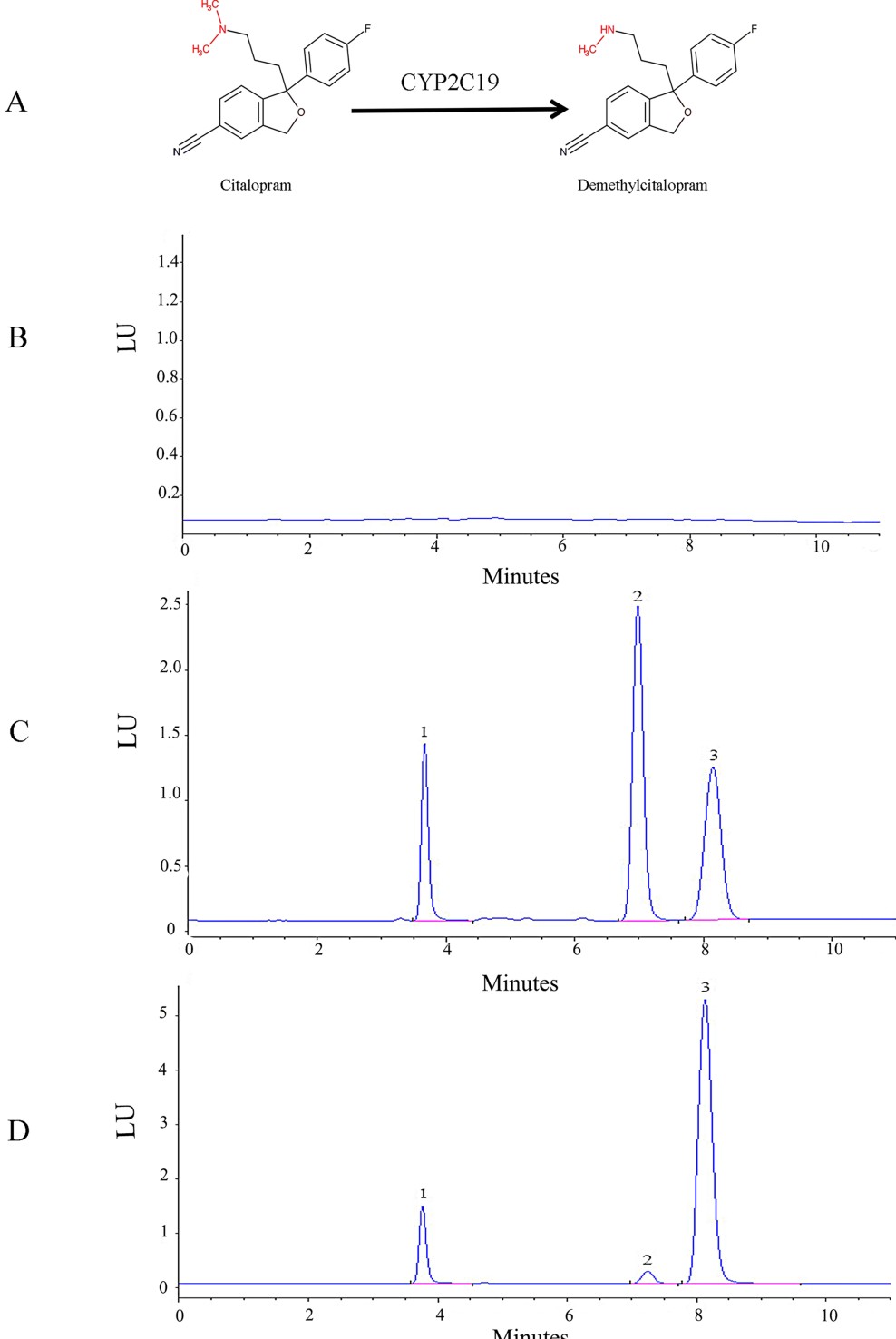

**Figure 1 Structural and HPLC characterization of citalopram metabolism by CYP2C19 in liver microsomes.** (A) Chemical structures and CYP2C19-mediated biotransformation pathway of citalopram; (B) HPLC chromatograms of the blank in recombinant liver microsomes; (C) HPLC chromatograms of the blank from inactivated recombinant liver microsomes containing standard drug solution–venlafaxine, demethylcitalopram and citalopram; (D) HPLC chromatograms of incubation sample of citalopram in recombinant liver microsomes.1: venlafaxine; 2: demethylcitalopram; 3: citalopram.

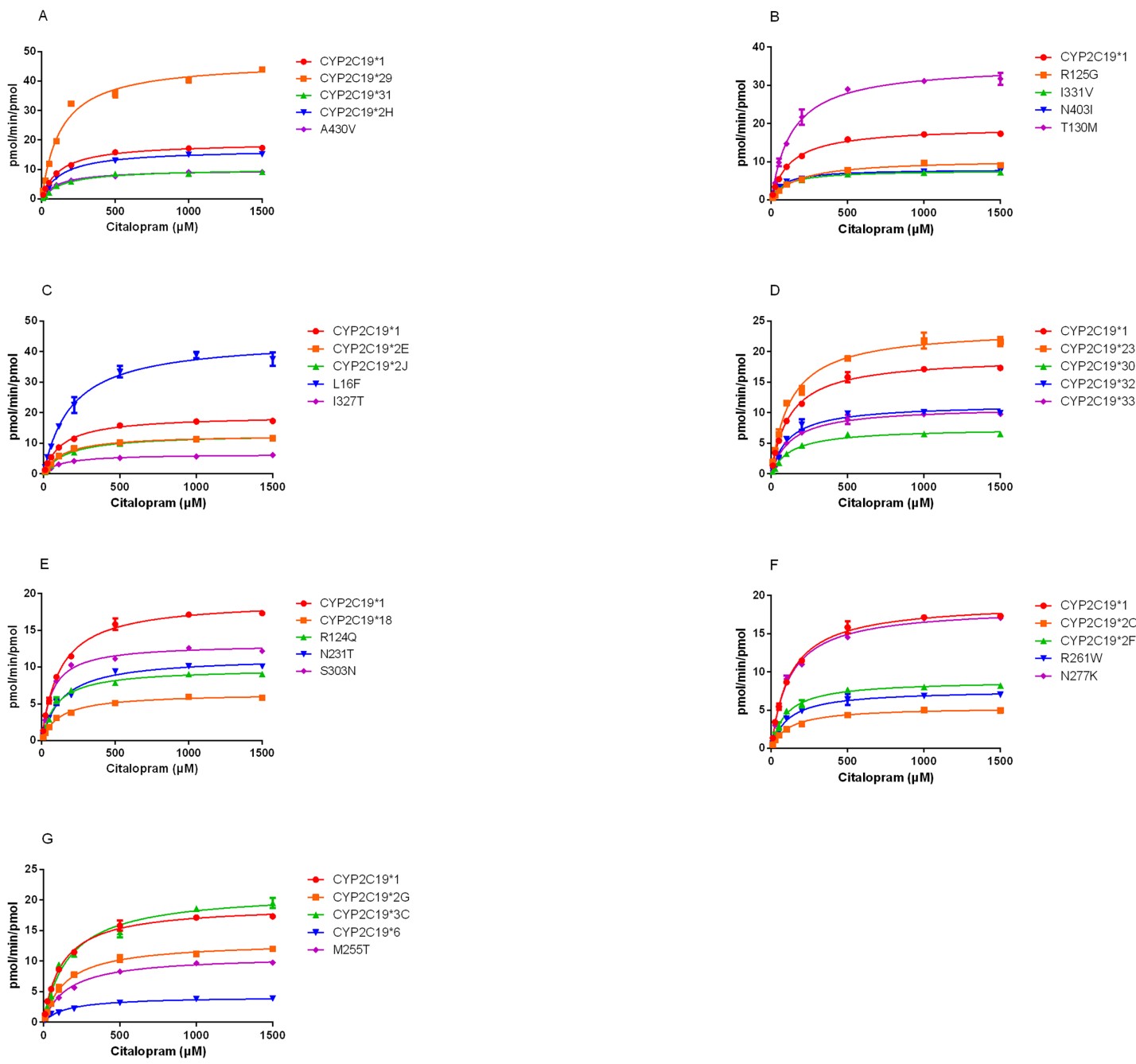

**Figure 2 Michaelis-Menten plots of kinetics formation for citalopram demethylation metabolism by recombinant wild-type and 30 CYP2C19 variants.** Each point corresponded to the Mean ± SD from three parallel samples. The variants have been manually arranged into seven different groups.

regarded as the ultrarapid metabolism variant. The remaining variants, which showed 70–130% of wild-type CLint, or no statistically significant difference in metabolic activity were classified as the normal metabolism variants.

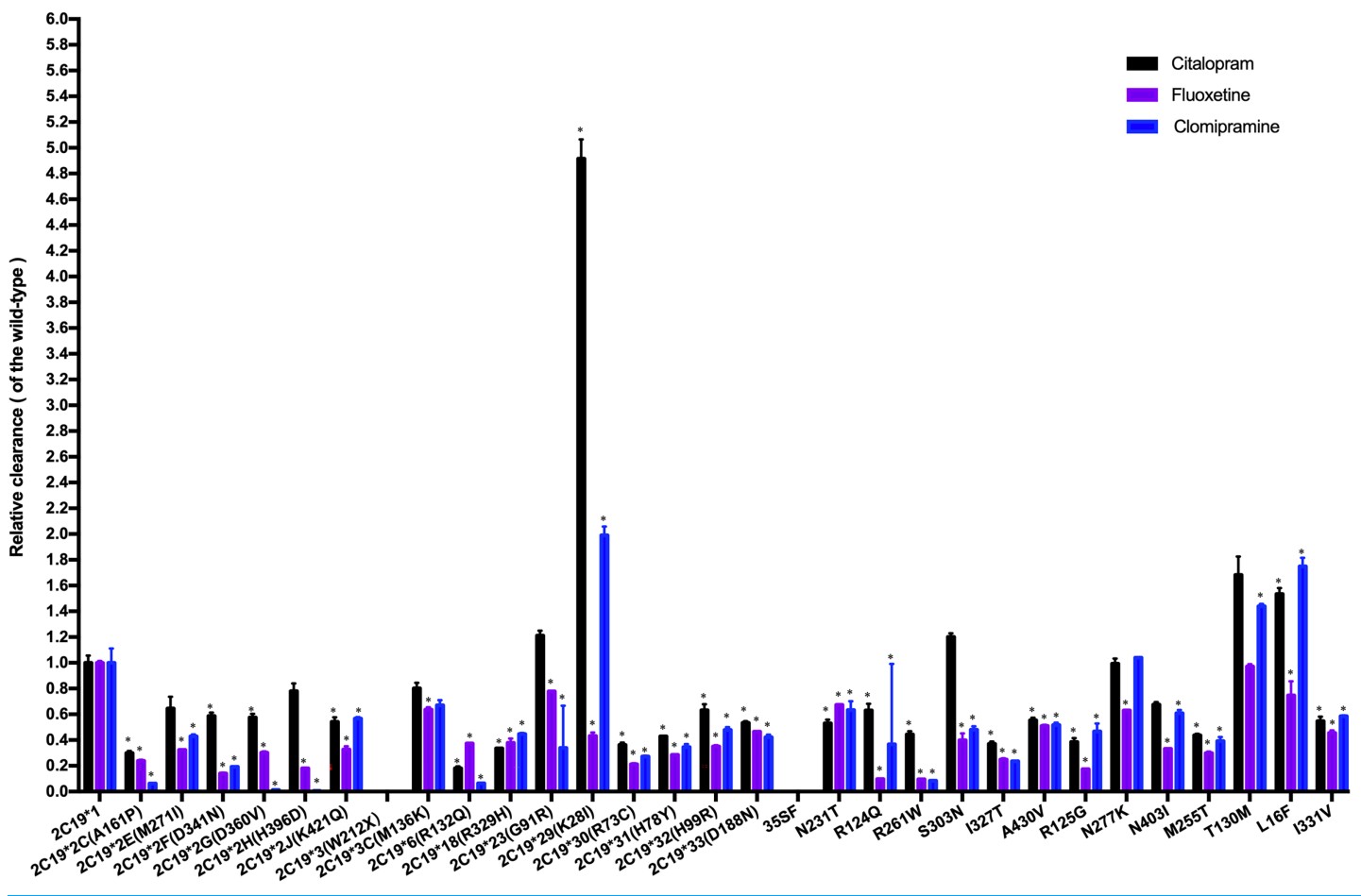

**Figure 3 Comparison of the metabolic activity of 30 CYP2C19 variants on citalopram, fluoxetine, and clomipramine.** *$P < 0.05$ compared with wild-type CYP2C19*1. Data are presented as the Mean ± SD of three independent experiments.

## DISCUSSION

The frequencies of some CYP2C19 variant alleles in our study are relatively low. However, research on these low-frequency variants remains clinically significant in the context of precision medicine, as these variants may still have an important effect on individualized treatments. Notably, there is a large population of 1.4 billion people in China. Therefore, even rare variants may have a large number of carriers. In addition, conducting clinical studies on these low-frequency CYP2C19 variants presents some challenges, such as difficulties in recruitment and the complexity of variable control. Therefore, this study employs an *in vitro* incubation system using recombinant human *CYP2C19* expressed in *Spodoptera frugiperda* (Sf) 21 insect cells to systematically evaluate the functional impact of 30 CYP2C19 variants on citalopram metabolism. Since *CYP2C19*1 is a high-frequency genotype and exhibits normal enzyme activity, it served as the control group. Finally, the results exhibited that most of the CYP2C19 variants significantly changed the kinetic parameters and influenced the metabolic activity of citalopram.

Based on our *in vitro* findings, we hypothesize that these *CYP2C19* variants may show similar enzyme activity *in vivo* and thus have potential implications for the individualized use of citalopram in clinical practice. Specifically, compared with individuals with normal metabolic activity, patients carrying poor or intermediate metabolizer variants may have slower metabolism of citalopram, potentially leading to increased drug concentrations and a higher risk of dose-related adverse reactions, such as neurotoxicity and cardiotoxicity. These individuals may require a slower titration process and lower maintenance doses. Conversely, patients carrying rapid or ultrarapid metabolism variants may metabolize citalopram more quickly, making it difficult to achieve therapeutic drug levels and resulting in inadequate efficacy. These patients may be considered for higher initial and maintenance doses. Nonetheless, these hypotheses require further validation through *in vivo* and real-world clinical studies.

CYP2C19 belongs to the subfamily CYP2C. Its gene is mapped to chromosome 10q23.33 and its enzyme is composed of 490 amino acids. The tertiary structure of CYP2C19 consists of 12 α-helices and three β-sheets. It exhibits two internal cavities: one cavity is positioned above the surface of the heme cofactor, where drug biotransformation occurs; and the other may form part of the substrate access/egress channel, which is connected to the active site (*Reynald et al., 2012*). The helices F, F', G', and G and their turns, the turn in β-hairpin 1, and the B-C loop region may be essential for the structural and functional stability of the CYP2C19 enzyme, and mutated residues in these regions may affect substrate binding and catalytic efficiency (*Reynald et al., 2012*; *Mustafa et al., 2019*). In addition, mutated residues located within the active pocket or substrate recognition sites (SRS) may also directly affect the enzyme's substrate affinity and catalytic efficiency. We hypothesize that these CYP2C19 variants showing significant changes in enzyme activity in our study are likely located in, or near, these functionally important regions.

*CYP2C19*\*2 and *CYP2C19*\*3 have been extensively studied in different ethnic populations as the most clinically significant defective alleles. *CYP2C19*\*3, which is most commonly found in East Asians and Oceanians, exhibits allele frequencies of 7% in East Asians and 15% in Oceanians (*Botton et al., 2021*). The loss of function of *CYP2C19*\*3 is caused primarily by a G to A transition at position 636 in exon 4, leading to a codon change from a tryptophan at position 212 to an early stop codon. This mutation produces a truncated protein missing the heme and substrate binding region, causing it to become nonfunctional. *CYP2C19*\*2 is most prevalent among Asians (29–35%) and African-Americans (15%) (*Scott et al., 2012*). In *CYP2C19*\*2, a synonymous G > A transition at position 681 in exon five creates an aberrant splice site, which alters the messenger RNA (mRNA) reading frame and results in the production of a truncated, nonfunctional protein. *In vitro* studies have shown that CYP2C19\*2 and \*3 variants almost completely lose their metabolic capacity due to impaired protein function, leading to little or no formation of the corresponding metabolites (*Dai et al., 2015*; *Lee et al., 2009*; *Shirasaka et al., 2016*). Our results also confirm that the CYP2C19\*3 variant is inactive, as no citalopram metabolites were detected. Similar to the CYP2C19\*3 variant, we observed that the 35FS variant also exhibits no enzymatic activity. The 35FS variant consists of the insertion of five nucleotides (CCTAC) at position 101 in exon 1 (*Hu et al., 2012*), which

may cause a frameshift and disruption to amino acid sequences, resulting in the production of an inactive protein. Furthermore, immunoblotting assays showed that CYP2C19 protein expression was absent for the allelic variants CYP2C19*3 and 35FS (*Dai et al., 2015*). These findings further indicate that *CYP2C19*3 and *35FS* are loss-of-function alleles. Research has shown that CYP2C19*2 and CYP2C19*3 variants can result in increased *in vivo* exposure to citalopram, thereby increasing the risk of intolerable side effects (*Chang et al., 2014*; *Wong et al., 2023*). According to CPIC guidelines, individuals carrying the *CYP2C19*3 or *CYP2C19*2 variants are classified as poor metabolizers who may need citalopram dose adjustments, such as a lower starting dose, a slower titration schedule, and a 50% reduction in the standard maintenance dose (*Bousman et al., 2023*). Given the same loss of enzyme activity observed in the 35FS variant, we speculate that patients carrying this mutation might also require similar dose adjustments. However, this hypothesis requires further validation in clinical research.

A published study on the structural characterization of CYP2C19 protein has found that residues within and adjacent to the helix B-C loop may play a critical role in substrate binding and enzyme activity (*Reynald et al., 2012*). In our study, we found six mutants, including CYP2C19*6 (R132Q), *30 (R73C), *31 (H78Y), *32 (H99R), R124Q, and R125G, whose mutation sites are located within or near the helix B-C loop. These variants showed a significant reduction in catalytic efficiency towards citalopram, ranging from 18.06% to 63.30% of wild-type. Our findings further support the notion that mutations in the helix B-C loop can impact CYP2C19 enzyme function. Characterization of the chimeric enzymes suggests that these mutations may alter the shape and chemical properties of the substrate-binding site by influencing the conformation and dynamics of the B-C loop. In addition, we found that residues R73, H99, R124, R125, and R132 are located in the reported SRS regions of CYP2C subfamily proteins (*Zawaira et al., 2011*). The SRS regions are involved in substrate recognition and binding, and contribute to positioning the substrate in the active site. Notably, Arg132 shows good conservation in the CYP2 family, and its positively charged side chain could stabilize the structure of the protein (*Lewis, 1998*). The R132Q variant may disrupt this stabilization, leading to altered enzyme activity. Furthermore, changes in the residues at position 125 in the CYP2C19 crystal structure have been reported to affect the active site and SRS 4 region (*Seo et al., 2023*). The R73C and H99R mutations may significantly influence the orientation and interactions of the CYP2C19-membrane system and affect the substrate access tunnels (*Mustafa et al., 2019*). These changes may impede substrate passage through the channel and slow down enzyme activity. Similar to our findings with citalopram, these six variants also showed impaired catalytic activity compared to wild-type enzyme towards mephenytoin and omeprazole (*Wang et al., 2011*; *Dai et al., 2015*). Therefore, the amino acid substitutions at key residues in the substrate recognition sites (SRS) regions of CYP2C19 may alter protein conformation and stability, and ultimately impair catalytic activity.

In addition, CYP2C19*2C, *18, R261W, I327T, and M255T variants demonstrated a dramatic decrease in CLint for citalopram in our study. Of these, CYP2C19*2C and *18 variants showed the most significant reductions in activity. Their CLint decreased to

approximately 30% of the wild-type enzyme level, mainly due to a comparable decrease in Vmax. The CYP2C19*2C variant carries an Ala161Pro substitution. Consistent with our findings, the Ala161Pro variant also exhibited only 30% of the wild-type enzyme activity for S-mephenytoin and omeprazole, two commonly used CYP2C19 probe substrates (*Wang et al., 2011*). These findings indicate that the Ala161Pro variant reduces catalytic efficiency. Sequence alignment analysis suggests that Ala161 is conserved in the human CYP2C subfamily (*Lewis, 2003*). It is located in the loop between the D-helix and the E-helix. It may contribute to stabilizing the helical structure of the D-E loop. Given the helix breaking property of proline, the Ala161Pro may change the conformation of the D-E loop and prevent substrate accessibility to the active site, thereby reducing the activity of the enzyme (*Wang et al., 2011*). However, a previous study reported that CYP2C19*18 (Arg329His) variant exhibited similar CLint compared to the wild-type enzyme when catalyzing S-mephenytoin and omeprazole, suggesting that this variant hardly alters metabolic activity for these substrates (*Wang et al., 2011*). This finding is in contradiction to our results. One possible reason is the substrate-dependent effect of this variant. The potential effects of Arg329His mutation on CYP2C19 enzymatic activity should be further studied. Moreover, the R261W, I327T, and M255T variants exhibited lower CLint than the wild-type enzyme (37.18-44.48%) for citalopram. Similar results were observed for both S-mephenytoin and omeprazole, where the value decreased by more than 50% (*Dai et al., 2015*). Our previous research has found that these variants show lower protein expression levels than that in the wild-type (*Dai et al., 2015*). We hypothesize that these substitutions interfere with the synthesis, folding, or stability of CYP2C19 protein by an uncertain mechanism, thus leading to the reduced catalytic activity.

CYP2C19*17 is one of the few known allelic variants associated with increased metabolic activity. It has been widely studied for its role in guiding the individualized use of citalopram. The enhanced activity of *CYP2C19*17 is primarily attributed to a C > T transition at the −806 site in the promoter region, which enhances the binding affinity for specific transcription factors, resulting in increased *CYP2C19* expression and activity (*Sim et al., 2006*). In this study, the CYP2C19*29 (K28I) and L16F variants also showed increased metabolic activity, with CYP2C19*29 showing particularly marked effects. Specifically, CYP2C19*29 exhibited approximately a 5-fold increase in CLint relative to wild-type, due to a 2.5-fold increase in Vmax and a 50% reduction in Km. However, unlike CYP2C19*17, which increases activity *via* increased protein expression, the CYP2C19*29 variant shows a relatively lower protein expression level than wild-type (*Dai et al., 2015*). This suggests that the protein expression level is not the only factor affecting enzyme activity. Structurally, the K28 residue is located in the linker region that connects the globular domain to the N-terminal transmembrane-helix. We speculate that the substitution of positively charged lysine at position 28 with hydrophobic isoleucine could change the polarity of the linker region. This change may significantly influence the orientation and interactions of CYP2C19 within the membrane environment (*Mustafa et al., 2019*). The optimized membrane orientation of CYP2C19 proteins may facilitate access of the hydrophobic drug citalopram to the active-site tunnels. This may be one possible reason for the observed increase in the metabolic activity of the K28I variant.

Clinically, individuals carrying the *CYP2C19*\*17 gain-of-function allele exhibit increased metabolism of citalopram to less active compounds when compared with normal metabolizers, leading to lower plasma concentrations and a potential reduction in clinical efficacy (*Chang et al., 2014*). Therefore, CPIC guidelines recommend that ultrarapid metabolizers be prescribed a higher maintenance dose (*Bousman et al., 2023*). If the gain-of-function effect of CYP2C19*29 is confirmed in clinical settings, individuals carrying this variant may similarly require careful monitoring of the therapeutic response to citalopram and dose individualization, akin to *CYP2C19*\*17 carriers.

Fluoxetine and clomipramine are commonly used antidepressants, metabolized *via* CYP2C19 just like citalopram. Previously, we employed the same *in vitro* incubation method to evaluate the metabolic impact of these 30 CYP2C19 variants on fluoxetine and clomipramine metabolism (*Fang et al., 2017*; *Lan et al., 2021*). For comparison, we combined the citalopram data with those of the other two antidepressants and analyzed the relative clearance of all three drugs across different CYP2C19 variants (Fig. 3). Our results indicate that most variants have a consistent trend of enzyme activity across the three antidepressants. For example, CYP2C19*3 and 35FS variants completely lacked enzymatic catalytic activity for all three antidepressants. However, eighteen CYP2C19 variants demonstrated significantly reduced CLint toward all three antidepressants compared to wild-type enzyme, with nine of them (CYP2C19*2C, *6, *18, *30, *31, R261W, I327T, R125G, and M255T) showing more than a 50% reduction. However, some variants (CYP2C19*23, *29, L16F, S303N, and T130M) exhibit different trends of enzyme activity among the antidepressants, especially CYP2C19*29 and L16F. The CYP2C19*29 variant displayed remarkable enhanced CLint toward citalopram (5-fold) and clomipramine (2-fold) compared with wild-type, yet showed reduced activity for fluoxetine (43.04% of the wild-type). Similarly, the L16F variant increased the CLint of citalopram and clomipramine to 1.5-fold and 1.7-fold relative to wild-type, respectively, while reducing fluoxetine metabolism to 74.14%. Mechanistically, we found that the same CYP2C19 variant can exhibit different metabolic tendencies depending on the substrate. Such substrate-specific metabolic behavior appears to be common among CYP2C19 variants. Structures and physicochemical differences of substrates may affect their binding modes and interactions with CYP2C19. Moreover, mutations may induce changes in enzyme conformation or structure, potentially reducing the enzyme's ability to accommodate certain substrates. These may collectively contribute to the substrate-dependent metabolic behavior of CYP2C19 variants (*Ibeanu et al., 1996*; *Derayea et al., 2019*). From a clinical perspective, these findings suggest that individuals carrying *CYP2C19*\*29 or *L16F* variants may need higher-than-standard therapeutic doses for clinical effectiveness with citalopram (or clomipramine), but lower-than-standard doses for avoiding toxic accumulation with fluoxetine. Therefore, clinicians should pay special attention to the differential metabolism of antidepressants caused by the same CYP2C19 variant, particularly during antidepressant switching.

Several limitations should be noted in our study. First, the functional characterization of the 30 CYP2C19 variants was assessed using a single substrate-citalopram. Therefore, the observed enzymatic activities should not be directly extrapolated to other substrates, as the

metabolic activity of CYP2C19 variants is known to be substrate-dependent. Second, although the *in-vitro* incubation method could efficiently and accurately assess the intrinsic metabolic capacity by minimizing confounding variables, it cannot completely simulate the complexity of the *in vivo* environment. Specifically, these *in vitro* systems may lack certain physiological cofactors and fail to reflect the actual expression levels of different variants. Moreover, most antidepressant drugs, such as citalopram and fluoxetine, are involved in metabolism through multiple pathways *in vivo*. We therefore remain interested in further validating the activity of variants *in vivo*. Third, we only constructed the key amino acid substitutions in the sixteen named variants, without introducing additional mutations found in complete haplotypes. Nevertheless, the current *in vitro* results can still provide a valuable reference for understanding their actual functions. In the future, *in vitro-in vivo* extrapolation (IVIVE) and pharmacokinetic modeling methods could be used to translate these *in vitro* results into clinical guidance.

## CONCLUSIONS

In this study, we functionally characterized the metabolic differences of 30 CYP2C19 variants in the N-demethylation of citalopram using an *in vitro* incubation system. Our findings showed that most of these variants exhibited significantly altered metabolic efficiency toward citalopram compared to the wild-type. These rare defective variants may partly contribute to interindividual variability in citalopram metabolism, which could potentially affect clinical response. Although most of the variants are not mentioned in pharmacogenetic guidelines, their potential functional effects are worth confirming through clinical and *in vivo* studies. Overall, these results provide foundational preclinical evidence for further genotype–phenotype correlation studies regarding the interindividual differences in citalopram metabolism.

## ACKNOWLEDGEMENTS

Wenzhou Medical University granted us access to laboratory facilities and equipment, which were essential for conducting our experiments. The Beijing Institute of Geriatrics of the National Health Commission generously donated critical experimental materials, including Cytochrome b5 microsomes and recombinant human CYP2C19 (expressed in *Spodoptera frugiperda* (Sf) 21 insect cells microsomes). We are grateful for their invaluable contributions to this study.

### Funding

This work was supported by the Science and Technology Research Plan of Jinhua City [grant number 2023-4-105]. The funders had no role in study design, data collection and analysis, decision to publish, or preparation of the manuscript.

## Grant Disclosures

The following grant information was disclosed by the authors:
Science and Technology Research Plan of Jinhua City: 2023-4-105.

## Competing Interests

The authors declare that they have no competing interests.

## Author Contributions

- Peng Wang conceived and designed the experiments, performed the experiments, analyzed the data, prepared figures and/or tables, authored or reviewed drafts of the article, and approved the final draft.
- Xiao-xia Hu performed the experiments, analyzed the data, prepared figures and/or tables, authored or reviewed drafts of the article, and approved the final draft.
- Jun-wei Li conceived and designed the experiments, analyzed the data, authored or reviewed drafts of the article, and approved the final draft.

## Data Availability

The raw measurements are available in the Supplemental File.

## Supplemental Information

Supplemental information for this article can be found online at http://dx.doi.org/10.7717/peerj.20027#supplemental-information.

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
