# Peer review of "In vitro assessment of the impact of 30 CYP2C19 variants on citalopram metabolism"

_PeerJ, doi:10.7717/peerj.20027_

## Round 0.1 · original submission · Major Revisions

The reviewers found your manuscript interesting, however they had a number of concerns that need to be addressed. They requested that you explain the rationale behind selecting the variants for your study and include specific key alleles in your study such as *17. Additionally, the figures and tables need to include information on statistical significance. The manuscript needs to explain how you controlled for the variability in enzyme expression in your system. Lastly, your discussion needs to address the limitations of your study as well as the clinical implications.

Please, submit a detailed rebuttal which shows where and how you have taken all comments and suggestions into consideration. If you do not agree with some of the reviewers’ comments or suggestions, please explain why. Your rebuttal will be critical in making a final decision on your manuscript. Please, note also that your revised version may enter a new round of review by the same or by different reviewers. Therefore, I cannot guarantee that your manuscript will eventually be accepted.

·

Basic reporting

The authors aim to test 30 CYP2C19 variants in an in vitro functional system, and refer to low frequency of alleles. It is unclear how they identified these variants. Reviewer requests adding a rationale for selection of these variants to the introduction. This rationale could include the reported frequencies for these variants.

The reviewer requests that the authors use either PharmVar nomenclature or amino acid change in the numbered codon, but not both. For PharmVar, haplotypes are defined, and
variants are sometimes included in haplotypes with other variants. For example, CYP2C19 *29 includes both K28I and I331V. Testing K28I separately from I331V should not be assigned *29. It is probably simplest to refer to the codon change (K281I) and then discuss how the findings may be relevant to the CYP2C19*29 allele.

Figure 3 is difficult to visualize variants with reduced Clint relative to wild-type. Suggest adding a column to Table 1 with ratio of Clint to wild-type.

Experimental design

The functional analysis of CYP2C19 variants, particularly rare variants, is an important advancement to the pharmacogenomics field. Reviewer recommends that the authors cite sources (such as PharmVar) that indicate specific alleles have uncertain function, such as CYP2C19 *29.

The SF21 insect system does not mimic liver microsomes. 'Liver' should be removed from the recombinant system description.

Validity of the findings

The limitations of the in vitro system should be described. For example, the system does not test for functional impact of reduced expression, which would also impact function. The in vitro system tests the same protein amount for each variant, but in vivo, variants may have different protein levels.

Fluoxetine is different from clomipramine and citalopram in terms of other metabolizing CYPs that may be involved in metabolism. Findings in a CYP2C19-only in vitro system are unlikely to reproduce in vivo metabolism, may not predict functional impact of CYP2C19 alleles on in vivo fluoxetine metabolism. This should be included in the discussion.

Reviewer 2 ·

Basic reporting

My comments are provided in the attachment, please review the attached review file.

Experimental design

My comments are provided in the attachment, please review the attached review file.

Validity of the findings

My comments are provided in the attachment, please review the attached review file.

Additional comments

My comments are provided in the attachment, please review the attached review file.

Annotated reviews are not available for download in order to protect the identity of reviewers who chose to remain anonymous.

---

## Round 0.2 · Minor Revisions

Thank you for thoroughly addressing the reviewers' comments and thus greatly improving your manuscript. One of the original reviewers reviewed your revised manuscript and suggested that you improve the grammar and language.

**Language Note:** The Academic Editor has identified that the English language must be improved. PeerJ can provide language editing services - please contact us at [email protected] for pricing (be sure to provide your manuscript number and title). Alternatively, you should make your own arrangements to improve the language quality and provide details in your response letter. – PeerJ Staff

Reviewer 2 ·

Basic reporting

Please see attached.

Experimental design

Please see attached.

Validity of the findings

Please see attached.

Additional comments

Please see attached.

Annotated reviews are not available for download in order to protect the identity of reviewers who chose to remain anonymous.

---

## Round 0.3 · Minor Revisions

Thank you for thoroughly addressing the reviewers’ comments and thus thoroughly improving your manuscript. One of the original reviewers re-reviewed your manuscript and suggested that you change the description of the variants in line 138 of your manuscript from novel to not well-studied or to having an unknown functional impact.

·

Basic reporting

Functional analysis of variation in CYP2C19 is important. Furthermore, this manuscript is much improved in terms of study rationale in the introduction. I also appreciate the addition of a Relative Clearance column in Table 1.

Line 138 indicates that the remaining 14 variants are novel. This is incorrect. I331V is present in many alleles, including CYP2C19 *1, *2, *3, *5, *11, *17. (https://www.pharmvar.org/gene/CYP2C19) Reviewer suggests describing these variants being not well-studied or having unknown functional impact, instead of novel.

Experimental design

No comment

Validity of the findings

No comment.

Additional comments

I recommend that the editor accept this manuscript for publication in PeerJ with very minor revisions.

---

## Round 0.4 · accepted · Accept

Thank you for addressing the reviewers' comments, thus improving your manuscript.

·

Basic reporting

Thank you for clarifying that many of the variants analyzed in this study have unknown functional impact.

Experimental design

no comment

Validity of the findings

no comment

Additional comments

no comment